# Investigation of Tunneling Effect for a N-Type Feedback Field-Effect Transistor

**DOI:** 10.3390/mi13081329

**Published:** 2022-08-16

**Authors:** Jong Hyeok Oh, Yun Seop Yu

**Affiliations:** ICT & Robotics Engineering, Semiconductor Convergence Engineering, AISPC Laboratory and IITC, Hankyong National University, 327 Jungang-ro, Anseong-si 17579, Gyenggi-do, Korea

**Keywords:** feedback field effect transistor, band-to-band tunneling, leakage current, tunneling length

## Abstract

In this paper, the tunneling effect for a N-type feedback field-effect transistor (NFBFET) was investigated. The NFBFET has highly doped N-P junction in the channel region. When drain-source voltage is applied at the NFBFET, the aligning between conduction band of N-region and valence band of P-region occur, and band-to-band tunneling (BTBT) current can be formed on surface region of N-P junction in the channel of the NFBFET. When the doping concentration of gated-channel region (*N_gc_*) is 4 × 10^18^ cm^−3^, the tunneling current makes off-currents increase approximately 10^4^ times. As gate-source voltage is applied to NFBFET, the tunneling rate decreases owing to reducing of aligned region between bands by stronger gate-field. Eventually, the tunneling currents are vanished at the BTBT vanishing point before threshold voltage. When *N_gc_* increase from 4 × 10^18^ to 6 × 10^18^, the tunneling current is generated not only at the surface region but also at the bulk region. Moreover, the tunneling length is shorter at the surface and bulk regions, and hence the leakage currents more increase. The BTBT vanishing point also increases due to increase of tunneling rates at surface and bulk region as *N*_gc_ increases.

## 1. Introduction

Recently, a metal oxide semiconductor field-effect transistor (MOSFET) has been challenged for limitation of switching speed and device scaling. Those challenges make hard to design next generation integrated circuits (ICs). In order to overcome these problem, novel devices, which have steep subthreshold slope (SS), were proposed, such as impact-ionization MOS (IMOS) [1,2,3], tunnel FET (TFET) [4,5,6,7], negative capacitance FET (NCFET) [8,9,10], and feedback FET (FBFET) [11,12].

Among them, the FBFET has been attracting attention as a next generation device, owing to complementary MOS (CMOS) based fabrication and high on/off ratio. The FBFET works on positive feedback, therefore, the FBFET has approximately zero subthreshold swing and hysteresis characteristics. Utilizing those characteristics, various circuits such as memory circuits, logic gates, bio-sensors, and neuromorphic circuits were presented [11,12,13,14,15,16,17,18,19,20,21,22,23,24,25,26,27,28,29,30]. Several structures of the FBFET have been proposed to meet the desired performance for each circuit. Most structures are based on S-shape energy band (or band-modulation) at the channel region. Mostly, the FBFET has P-N-P-N and P-N-i-N structure, and the family of Z^2^-FET, which is works on same mechanism, has P-i-N. The S-shape energy band consists of potential barrier and well. The potential barrier controls carrier injection at gated channel region. The carriers are injected by thermal emission to adjust the gate-source voltage. Then, the carriers are accumulated at ungated channel region. The accumulated carriers eliminate the potential well, and finally the carriers flow into the other side contact by diffusion. These mechanisms make the positive feedback between electron and hole in the channel region, and therefore the energy band of all regions is aligned. When the potential barrier and well are not high enough to make S-shape energy band, the FBFET works like a diode. Therefore, to make positive feedback in the channel region, highly doped region or virtually highly doped region are required. When a drain-source voltage is applied to the FBFET, the valence band of potential barrier and the conduction band of potential well were aligned, and the band-to-band tunneling (BTBT) current can be generated. The BTBT tunneling current make potential barrier and well lower, and therefore the leakage current increases and the threshold voltage changes. So far, when investigating the electrical characteristics of the FBFET using technology computer aided design (TCAD), physical models of MOSFET and bipolar junction transistor (BJT) were used according to FBFET structure. Most studies were not considered tunneling current in the channel region. To accurately investigate the electrical characteristics of the FBFET, it is necessary to consider the tunneling current that can have occurred at the channel region.

In this paper, the tunneling effect in the N-type FBFET(NFBFET) is investigated with various channel doping profiles. First, the energy band diagram for describing the simulation condition and the mechanism of FBFET will be introduced in Section 2. Then, simulation results of the electrical characteristics considering tunneling effect of the FBFET will be discussed in Section 3. Finally, conclusion will be described.

## 2. Simulation Results

### 2.1. Simulation Structure and Parameter

Figure 1 shows a 2-dimensional (2D) schematic diagram of NFBFET. The P-N-P-N structure was used for simulation, and the NFBFET was simulated with commercial simulator ATLAS by Silvaco [31]. Table 1 shows the structure parameter of the NFBFET. The doping concentration of gated-channel regions was variable. The material of the gate-oxide is aluminum oxide (Al_2_O_3_), and the work-function of the gate is 5.0 eV. For investigating the electrical characteristics for the NFBFET, the physical models of MOSFET and BJT were used. Those models include the transverse field dependent mobility model (CVT), field-dependent mobility (FLDMOB), Shockley-Read-Hall recombination model (SRH), Auger recombination model (AUGER), bandgap narrowing model (BGN), and Fermi-Dirac calculation model (FERMI). Additionally, for considering the tunneling effect, the non-local BTBT model (BBT.NONLOCAL) was used. The simulation was conducted by transient analysis, and the long term of time step (~1 s) was used to investigate DC characteristics of the NFBFET.

### 2.2. Mechanism of the NFBFET

Figure 2 shows the energy band diagram for describing the mechanism of the NFBFET. In this section, tunneling effect was not considered. For this structure, *N_gc_* is 4 × 10^18^ cm^−3^. The black and red lines denote valence band and conduction band, respectively. Figure 2a shows the initial state of energy band diagram for NFBFET. For this state, the potential barrier formed by drain-source junction blocks carrier injection. When drain-source voltage (*V_DS_*) is applied to the NFBFET at 1 V, as shown in Figure 2b, carriers are injected into the channel region. When the gate-source voltage (*V_GS_*) is applied to the NFBFET with the forward sweep, the potential barrier is lower. The electron from source region can inject into channel region by thermal emission. The injected electrons accumulate at the potential well. As the *V_GS_* increases, the accumulated electrons increase, and then the potential well is eliminated. The hole from the drain region can inject into the channel region, as shown in Figure 2c. Finally, the conduction and valance bands of all regions are aligned, as shown in Figure 2d. The positive feedback occurs at threshold voltage accelerating the lowering barrier and well. Figure 3 shows the drain-source current-gate-source voltage (*I_DS_*-*V_GS_*) characteristics at *V_DS_* = 1 V. In the subthreshold region, off-current is made by minority carrier diffusion at drain-side junction. While the *V_GS_* increases, the leakage current increases by lowered potential well. Finally, the current increases rapidly by accelerating positive feedback near the threshold voltage, as shown in Figure 3.

## 3. Tunneling Effect for the NFBFET

The NFBFET has highly doped N-P junction in the channel region. When the *V_DS_* is applied to NFBFET such that the conduction band of the N-region and the valence band of the P-region are aligned, the BTBT can occur at N-P junction in the channel region, as shown in Figure 2b. Then, the BTBT currents make the carrier concentration in channel region change. Therefore, it is necessary to investigate the tunneling effect in the channel region. 

Figure 4 shows *I_DS_*-*V_GS_* characteristics of NFBFET at *V_DS_* = 1 V when the BTBT is considered and ignored. The solid and dash lines denote simulation results considering and ignoring the BTBT, respectively. The leakage current considering BTBT increases about 10^4^ times compared to ignoring BTBT when *V_GS_* = 0 V. As *V_GS_* increases, the conduction and valance bands of the gated channel region decrease, and then the BTBT current decreases due to increase of tunneling length at N-P junction in the channel region.

### 3.1. V_DS_ Dependence of Leakage Current 

Figure 5a,b show the electron tunneling rate of the NFBFET at two cases of *V_DS_* and fixed *V_GS_* = 0 V when the BTBT is considered. The black and red square-lines denote tunneling rates on bulk and surface regions, respectively. For the initial state (*V_DS_* = 0 V), there is no BTBT, as shown in Figure 5a. When *V_DS_* is applied at 1 V, the tunneling rate increase to approximately 10^24^ cm^−3^s^−1^, as shown in Figure 5b. Moreover, the BTBT occurs on the surface near N-P junction in the channel region due to band-bending by gate work-function difference in metal-oxide-semiconductor structure.

Figure 6a,b show the carrier concentrations on surface region of the NFBFET at two cases of *V_DS_*s and fixed *V_GS_* = 0 V when the BTBT is considered. The red and black lines denote electron and hole concentrations, respectively. The solid and dash lines denote concentrations with BTBT and non-BTBT, respectively. The carrier concentration of initial state is determined by doping concentration due to no BTBT currents, as shown in Figure 6a. The changed carrier concentrations at *V_DS_* = 1 V are shown in Figure 6b. When *V_DS_* increases, BTBT occurs on the surface of channel region. Excess carriers, which are generated by BTBT, are hard to flow into the electrode, owing to high potential barriers of drain and source-side junctions. Then, generated carriers accumulate in the channel region. Subsequently, accumulated carriers make the potential barrier and well of the channel region lower. Because lowered potential barrier permit injection from the source by thermal emission, the leakage current occurs. This mechanism is similar to the positive feedback. 

Figure 7 show the energy bands on surface region at *V_GS_* = 0 and *V_DS_* = 1 V when BTBT is considered and ignored. The red and black lines denote conduction and valence bands, respectively. As shown in Figure 7, the potential barrier and well is shifted by changing carrier concentrations in the channel region. The shallowed potential well produces more minority carrier diffusion current at *V_GS_* = 0 V. Therefore, the leakage currents increase when the BTBT is considered, as shown in Figure 4.

### 3.2. V_GS_ Dependence of Leakage Current

Figure 8a,b show the electron tunneling rates and energy band on surface region at *V_GS_* = 0.15 to 0.45 V with bias step of 0.1 V and *V_DS_* = 1 V. The black, red, green, and blue lines denote the electron tunneling rates and energy band at *V_GS_* = 0.15, 0.25, 0.35, and 0.45 V, respectively. The non-local tunneling process can be calculated with local tunneling process by assuming that the electric field is uniform at tunneling path [32]. When tunneling current with local process, the tunneling rate is related to tunneling length. For the Kane model, which is representative local model, the tunneling rate *G*_tun_ for BTBT is given by [33,34,35]
(1)$$G_{\mathrm{tun}}=\frac{A_{k}}{\sqrt{E_{g}}}{\left(\frac{E_{g}}{qW_{t}}\right)}^{2.5}\exp\left(-qB_{k}W_{t}\sqrt{E_{g}}\right),$$
where *A*_k_ and *B*_k_ are nonlocal BTBT model parameter, and *E*_g_ and *W*_t_ are energy gap and tunneling length, respectively. The tunneling rate has an exponential relation with tunneling length. As *V_GS_* increases from 0.15 to 0.35 V, the tunneling rates decrease exponentially, owing to increase of tunneling length between conduction band of N-region and valence band of P-region by gate-field, as shown in Figure 8b. They reach zero approximately at *V_GS_* = 0.45 V, as shown in Figure 8a. Moreover, as the tunneling rates decrease, the electrons injected into potential well by BTBT decrease. Eventually, the change of potential well by BTBT is vanished, as shown in Figure 8b. Accordingly, the BTBT currents reduce as shown in Figure 4.

Figure 9a,b show the carrier concentrations on surface region at *V_GS_* = 0.35 and 0.45 V when BTBT is considered and ignored. The red and black lines denote electron and hole concentrations, respectively, and the solid and dash lines denote electron and hole concentrations with BTBT and non-BTBT, respectively. As shown in Figure 9a, carrier concentrations considering BTBT at *V_GS_* = 0.35 V are still some different from those ignoring BTBT, owing to remaining of BTBT at the surface. However, there is no longer a difference between carrier concentrations considering and ignoring BTBT when BTBT disappear at *V_GS_* = 0.45 V, as shown in Figure 9b. Therefore, as *V_GS_* increases, the current decreases, as shown in Figure 4. When *V_GS_* > 0.45 V, the current considering and ignoring BTBT are the same since BTBT does not exist, as shown in Figure 4.

### 3.3. Tunneling Effect with Various Doping Profile

Figure 10a shows the *I_DS_*-*V_GS_* characteristics of the NFBFET considering BTBT with *N_gc_* from 4 × 10^18^ to 6 × 10^18^ cm^−3^ at *V_DS_* = 1 V. The filled and empty symbols denote the simulation results for considering and ignoring BTBT, respectively. As *N_gc_* increases, the leakage current increases. When *N_gc_* increase, the tunneling length is shorter, as shown in Figure 10b. Hence, the tunneling rates increase according to Equation (1). Moreover, higher *N_gc_* make tunneling rates higher on surface as well as bulk region of the NFBFET, as shown in Figure 10c. However, as *V_GS_* increases, the BTBT currents decreases by lowering potential barrier on gated-channel region with stronger gate-field, and eventually they are vanished at BTBT vanishing points, as shown in Figure 10a, and then thermionic emission mechanism becomes dominant, such as the FBFET ignoring BTBT. The BTBT vanishing point increases due to increase of tunneling rates on surface and bulk region as *N*_gc_ increases.

## 4. Conclusions

The tunneling effect of the NFBFET was investigated in the case of *N_gc_* of from 4 × 10^18^ to 6 × 10^18^ cm^−3^. First, when *N_gc_* is 4 × 10^18^, the BTBT currents is generated on the surface region of the NFBFET. This BTBT currents make off-current increase about 10^4^ times. As *V_GS_* increases, the BTBT currents decrease by lowering potential barriers on the gated-channel region with stronger gate-field, and eventually BTBT currents are vanished, and then thermionic emission currents become dominant such as the FBFET ignoring BTBT. As *N_gc_* increase from 4 × 10^18^ to 6 × 10^18^ cm^−3^, the tunneling length is shorter. Following the tunneling probability equation, tunneling rate increases as *N_gc_* increases. Accordingly, the leakage currents increase due to increase of tunneling rate. Moreover, the BTBT currents can be generated on surface region as well as bulk region. As *V_GS_* increases, the BTBT currents are also vanished before the threshold voltage, and the BTBT vanishing point increase as *N*_gc_ increases. According to these results, it is necessary to further investigate the tunneling effect of NFBFET on a very high doping channel region.

## Figures and Tables

**Figure 1 micromachines-13-01329-f001:**
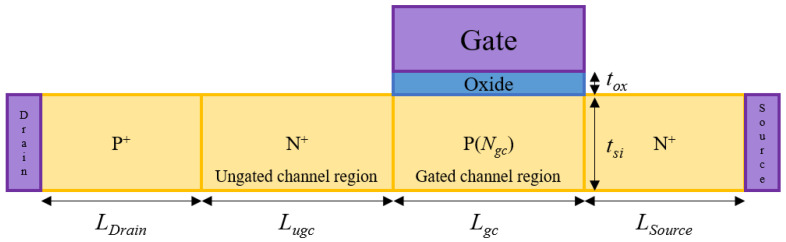
2D schematic diagram of the NFBFET.

**Figure 2 micromachines-13-01329-f002:**
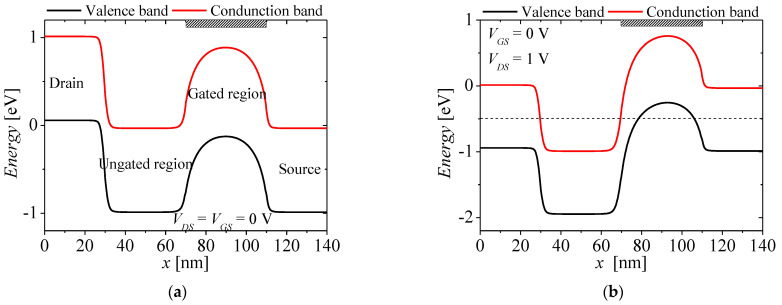
Energy band diagrams of NFBFET of each state. (**a**) initial state (*V_DS_* = *V_GS_* = 0 V), (**b**) *V_GS_* = 0 V and *V_DS_* = 1 V, (**c**) forward sweep of gate−source voltage *V_GS_* at *V_DS_* = 1 V, (**d**) on−state by *V_GS_* at *V_DS_* = 1 V.

**Figure 3 micromachines-13-01329-f003:**
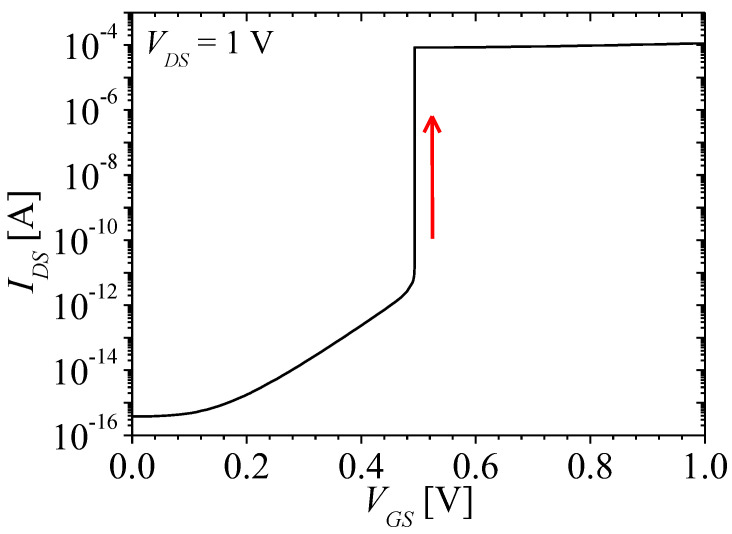
Drain−source current − gate−source voltage (*I_DS_*-*V_GS_*) characteristics of NFBFET at *V_DS_* = 1 V.

**Figure 4 micromachines-13-01329-f004:**
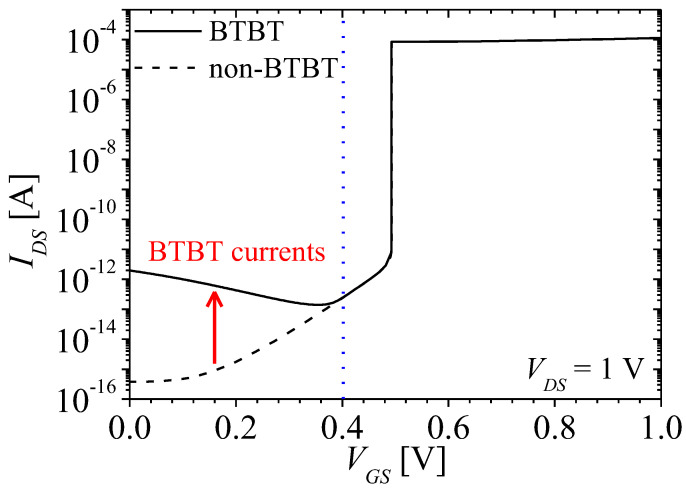
Comparison of *I_DS_*-*V_GS_* characteristics of NFBFET between considering and ignoring the BTBT at *V_DS_* = 1 V.

**Figure 5 micromachines-13-01329-f005:**
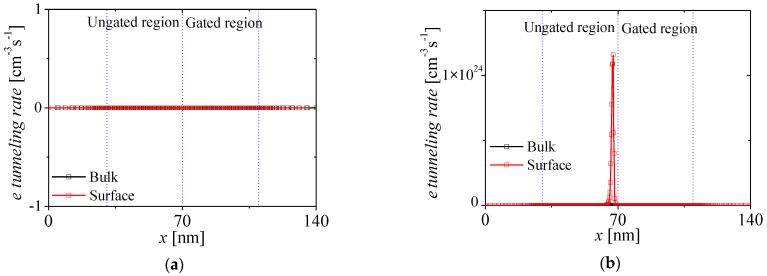
Electron tunneling rates at two cases of *V_DS_* on bulk and surface regions. (**a**) *V_DS_* = 0 V (initial state) and (**b**) *V_DS_* = 1 V.

**Figure 6 micromachines-13-01329-f006:**
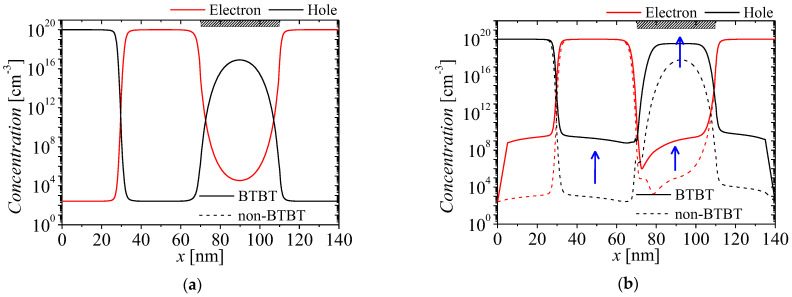
Comparison of carrier concentrations on surface region at two cases of *V_DS_*. (**a**) *V_DS_* = 0 V (initial state) and (**b**) *V_DS_* = 1.0 V.

**Figure 7 micromachines-13-01329-f007:**
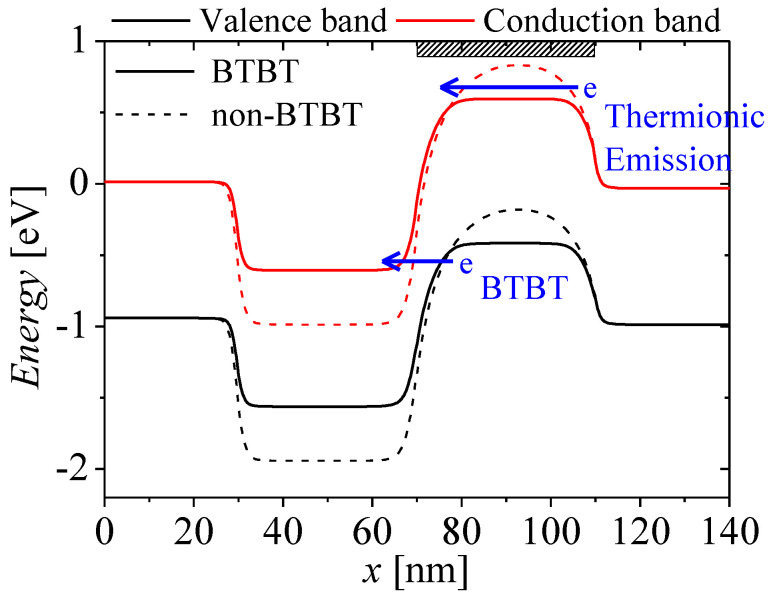
Comparison of energy bands on surface region at *V_GS_* = 1 V and *V_DS_* = 1 V when BTBT is considered and ignored.

**Figure 8 micromachines-13-01329-f008:**
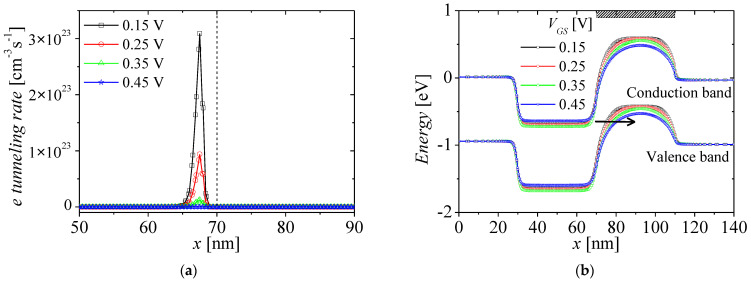
(**a**) Electron tunneling rates and (**b**) energy bands on surface region at *V_GS_* = 0.15 (black), 0.25 (red), 0.35 (green), and 0.45 V (blue) at *V_DS_* = 1 V.

**Figure 9 micromachines-13-01329-f009:**
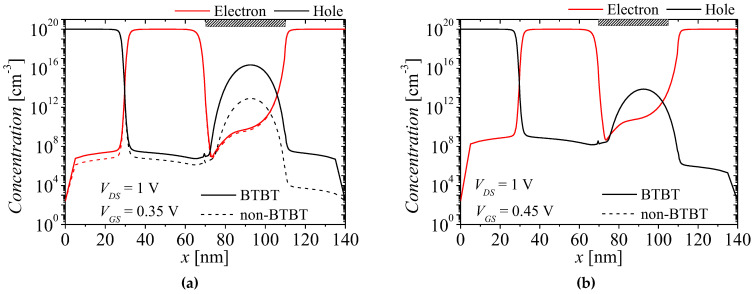
Comparison of carrier concentrations on surface region when BTBT is considered and ignored at *V_DS_* = 1 V. (**a**) *V_GS_* = 0.35 V and (**b**) *V_GS_* = 0.45 V.

**Figure 10 micromachines-13-01329-f010:**
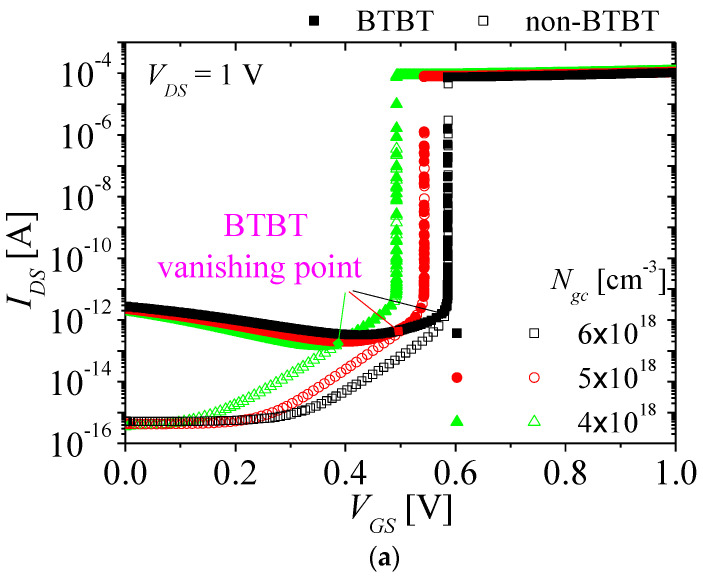
(**a**) Comparison of *I_DS_*-*V_GS_* characteristics of the NFBFET with *N_gc_* from 4 × 10^18^ to 6 × 10^18^ between considering and ignoring the BTBT, (**b**) the energy band on the surface region for ignoring the BTBT near N−P junction in the channel region with *N_gc_* from 4 × 10^18^ to 6 × 10^18^ at *V_DS_* = 1 V and *V_GS_* = 0 V, and (**c**) the tunneling rates from surface to bulk region of the NFBFET with *N_gc_* from 4 × 10^18^ to 6 × 10^18^ at *V_DS_* = 1 V and *V_GS_* = 0 V.

**Table 1 micromachines-13-01329-t001:** Structure parameter of the NFBFET for TCAD simulation.

Parameters	Description	Value/Unit
*L_drain_*	Length of drain region	30 nm
*L_ugc_*	Length of ungated channel region	40 nm
*L_ugc_*	Length of gated channel region	40 nm
*L_source_*	Length of source region	30 nm
*T_si_*	Thickness of silicon body	15 nm
*T_ox_*	Thickness of gate-oxide	3 nm
*N_gc_*	Doping concentration of gated channel region	Var.
	Doping concentration of P^+^ region	1 × 10^20^ cm^−3^
	Doping concentration of N^+^ region	1 × 10^20^ cm^−3^

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
