# Peer review of "Investigation of Tunneling Effect for a N-Type Feedback Field-Effect Transistor"

_micromachines, 2022, doi:10.3390/mi13081329_

Round 1

Reviewer 1 Report

Comments and suggestions:

Manuscript title: “Investigation of Tunnelling Effect for a N-type Feedback Field- 2 effect Transistor”

Publisher: Micromachines.

Comments a suggestion: The   authors   the manuscript “Investigation of Tunnelling Effect for a N-type Feedback Field- 2 effect Transistor” have used very good scientific language. Explanation and   figures quality are also very good. Over all the prepared manuscript is adoptable for publication. Some general quarries and suggestion are listed below. Hope the quality of submitted manuscript will improve more, if authors include suggestions:

1: There are some typos and grammatical errors in the manuscript.  Revise   manuscript carefully.  Remove errors.

2:  Figures/results visibility   is poor.   Re- edit all figures/results   carefully by using   appropriate colour and font size.

3:  Introduction part seems little bit weak, re edit introduction part by using   appropriate and  sufficient  number of references.

  • A. M. Ionescu and H. Riel, “Tunnel field-effect transistors as energy efficient electronic switches,” Nature, Vol. 479, No. 7373, pp. 329–337, Nov. 2011.
  • T. Krishna Mohan, D. Kim, S. Raghunathan, and K. Saraswat, “Double-gate strained-Ge heterostructure tunneling FET (TFET) with record high drive currents and <60mV/dec subthreshold slope,” Proc. IEEE Int. Electron Devices Meeting (IEDM), Dec. 2008, pp. 1–3.
  • J. T. Smith, S. Das, and J. Appenzeller, “Broken-gap tunnel MOSFET: A constant-slope sub-60-mV/decade transistor,” IEEE Electron Device Letter, Vol. 32, No. 10, pp. 1367–1369, Oct. 2011.
  • S. B. Rahi, S. Tayal, A. Kumar, “Emerging Negative Capacitance Field Effect Transistor in Low Power Electronics”, Microelectronics Journal Volume 116,2021, 105242, https://doi.org/10.1016/j.mejo.2021.105242
  • E. Yu, L. Wang, Y. Taur, and P. Asbeck, “Design of tunneling field-effect transistors based on staggered heterojunctions for ultralow-power applications,” IEEE Electron Device Letters, Vol. 31, No. 5, pp. 431–433, May 2010.
  • 4:     Tunnel FET is a steep subthreshold slope FET catrgory.   Subthreshold Slope is an important design feature.   Author should elaborate its subthreshold slope   feature.

Author Response

Please find an attached file.

Reviewer 2 Report

The authors have submitted a timely work which is based on a feedback FET. I would recommend the work given that the authors clarifies a few major concerns. Kindly specify the used materials for source/drain and gate metals. Are all the contacts considered ohmic? Is the use of Al2O3 appropriate for direct deposition over silicon? Have the authors considered the possibility of trap charges? As the device length is greater than 100nm, why the authors neglected the impact ionization even with such high possibility of critical electric field at the turn-on bias? Is the device calibrated with experimental setup? if yes, then mention the calibrated parameter (transit time, tau)? As the author mentioned initially that non-local BTBT is used for the simulation but later on kane method was considered, why (which is local in case of silvaco)? Kindly mentioned the used Ak, Bk, Eg, and Wt parameters for clarity of the reader. Why the concentration increases steeply around 70nm in Fig. 9(b)? The proposed device should be compared with other steep-subthreshold device to show the merits. It would be convenient to present the used code for the simulation either as a supplementary material or in review comments to verify if the results are reproducible.

Author Response

Please find an attached file.

Round 2

Reviewer 2 Report

The authors have clarified most of the concerns. I believe the IDS-VGS characteristics are extracted by joining every single IDS value for transient analysis with a constant Vds and Vgs. For the last comment, I would recommend the authors to clarify how the IDS-VGS are extracted and possible options to extract the IDS-VDS in the menuscript by explaining the solve statement in the manuscript.

Author Response

Please find an attached file.

Yun Seop Yu
